# Ladostigil Reduces the Adenoside Triphosphate/Lipopolysaccharide-Induced Secretion of Pro-Inflammatory Cytokines from Microglia and Modulate-Immune Regulators, TNFAIP3, and EGR1

**DOI:** 10.3390/biom14010112

**Published:** 2024-01-16

**Authors:** Fanny Reichert, Keren Zohar, Elyad Lezmi, Tsiona Eliyahu, Shlomo Rotshenker, Michal Linial, Marta Weinstock

**Affiliations:** 1Department of Medical Neurobiology, Institute for Medical Research Israel-Canada (IMRIC), Faculty of Medicine, Hebrew University of Jerusalem, Jerusalem 91120, Israel; funarei@gmail.com (F.R.); shlomor@ekmd.huji.ac.il (S.R.); 2Department of Biological Chemistry, Institute of Life Sciences, Hebrew University of Jerusalem, Jerusalem 91904, Israel; keren.zohar@mail.huji.ac.il (K.Z.); tsiona.e@mail.huji.ac.il (T.E.); michall@cc.huji.ac.il (M.L.); 3Department of Genetics, Institute of Life Sciences, Hebrew University of Jerusalem, Jerusalem 91904, Israel; elyad.lezmi@mail.huji.ac.il; 4Institute of Drug Research, School of Pharmacy, Hebrew University of Jerusalem, Jerusalem 91120, Israel

**Keywords:** aging rats, NFκB, NLRP3 inflammasome, P2x7 receptor, primary murine microglia, RNA-seq

## Abstract

Treatment of aging rats for 6 months with ladostigil (1 mg/kg/day) prevented a decline in recognition and spatial memory and suppressed the overexpression of gene-encoding pro-inflammatory cytokines, TNFα, IL1β, and IL6 in the brain and microglial cultures. Primary cultures of mouse microglia stimulated by lipopolysaccharides (LPS, 0.75 µg/mL) and benzoyl ATPs (BzATP) were used to determine the concentration of ladostigil that reduces the secretion of these cytokine proteins. Ladostigil (1 × 10^−11^ M), a concentration compatible with the blood of aging rats in, prevented memory decline and reduced secretion of IL1β and IL6 by ≈50%. RNA sequencing analysis showed that BzATP/LPS upregulated 25 genes, including early-growth response protein 1, (Egr1) which increased in the brain of subjects with neurodegenerative diseases. Ladostigil significantly decreased Egr1 gene expression and levels of the protein in the nucleus and increased TNF alpha-induced protein 3 (TNFaIP3), which suppresses cytokine release, in the microglial cytoplasm. Restoration of the aberrant signaling of these proteins in ATP/LPS-activated microglia in vivo might explain the prevention by ladostigil of the morphological and inflammatory changes in the brain of aging rats.

## 1. Introduction

Microglia are the resident immune cells in the brain. In a healthy adult brain, microglia have a ramified morphology and long processes [1]. They are involved in the integration of new neurons into neuronal circuits, which is important for learning, memory, and cognition [2]. In response to injury, microglia contract their processes, assume an amoeboid shape, and proliferate and migrate toward the site of injury. ATP, released from injured neurons, stimulates purinergic receptors on the microglial membrane, triggering an efflux of K^+^ that activates the nucleotide-binding oligomerization domain-(NOD)-LRR and pyrin domain inflammasome (NLRP3) and converts procaspase-1 to caspase-1 [3]. This enables the processing and secretion of IL1β and other pro-inflammatory cytokines [4].

Microglia have retracted processes in the aging brain [5] in which cell damage results from a decline in mitochondrial activity and antioxidant defense mechanisms [6,7]. They also have higher levels of pro-inflammatory cytokines and express cytokine receptors, which could contribute to neurodegeneration and memory impairment [8,9].

Basal levels of TNFα are necessary for regulating synaptic transmission and plasticity [10], while IL1β levels are needed to regulate long-term potentiation (LTP) that underlies learning and memory [11]. However, excess amounts of these cytokines can impair these cellular processes [12], as demonstrated by the direct injection of IL1β into the brain, which inhibits hippocampal LTP [13,14]. Also, greater levels of brain IL6 are linked to synapse loss and deficits in avoidance learning in mice [15].

Oxidative stress and cytokines activate signaling pathways, such as the mitogen-activated protein kinase (MAPK) family of proteins in immune cells [16]. MAPKs consist of three main families: extracellular signal-regulated kinase (ERK), c-Jun N-terminal kinase (JNK), and p38 [17]. IL1β activates MAPK p38 and JNK and increases the nuclear factor kappa-light-chain-enhancer of activated B cells (NFκB) [18], which is strongly associated with age in mice and humans [19]. NFκB is increased in the brains of subjects with neurodegenerative diseases [20]. This finding prompted a search for novel therapies to slow age-related memory impairment by inhibiting the nuclear translocation of NFκB [21].

For this purpose, we developed ladostigil (6-(N-ethyl, N-methyl carbamyloxy)-N propargyl-1(R)-aminoindan hemitartrate), which significantly reduced the mitochondrial potential in cells subjected to oxidative–nitrative stress and decreased malonaldehyde, a measure of oxidative stress, in the cerebral hemispheres of mice, induced by the injection of a lipopolysaccharide (LPS) [22] (Panarsky, 2012). In primary mouse microglial cultures activated by LPS, ladostigil reduced the nuclear translocation of NFκB and phosphorylation of ERK1/2 and p38 and downregulated the gene expression of TNFα, IL6, and IL1β [23]. In addition, when ladostigil was administered at a dose of 1 mg/kg/day for 6 months to 16-month-old rats, it prevented a decline in recognition and spatial memory [24]. It also suppressed the increase in mRNA of TNFα, IL6, and IL1β induced by aging [23] (Panarsky et al., 2012) and genes adversely affected by synaptic function in brain regions associated with learning and memory [25].

The aim of the current study was to obtain a better understanding of the mechanism through which ladostigil reduces cytokine release from microglia. The preparation of such primary microglial cultures causes the loss of some membrane receptors that are activated in the intact brain [26]. However, they still retain purinergic receptors for adenosine and ATP, released in response to neuronal injury [27]. The activation of the (P2x7R) subtype results in the processing and secretion of TNFα and IL1β in response to LPS [28]. The secretion of IL1β can be achieved by adding 2′-3′-O-(4-benzoyl benzoyl) adenosine 5′-triphosphate (BzATP), an agonist of P2x7R [29].

We first looked for the concentrations of ladostigil that would maximally inhibit the secretion of IL1β, IL6, and TNFα proteins from microglia activated by a combination of BzATP and LPS. Then, we measured the levels of ladostigil in the blood of the aging rats in which a dose of 1 mg/kg/day had prevented memory decline to ascertain whether there was enough of the drug in vivo to have affected cytokine release from microglia. Lastly, we sought additional information about the cellular processes involved in these actions of ladostigil, using RNA sequencing (RNA-seq) to perform a detailed analysis of its effect on gene expression in the microglia.

## 2. Materials and Methods

### 2.1. Animals

Male Balb/C mice and Wistar rats purchased from Harlan Laboratories (Jerusalem, Israel) were used in accordance with the National Research Council’s guide for the care and use of laboratory animals. The Animal Care and Use Committee of the Hebrew University approval #MD-19-15710-4 was for the mice, #MD-08-11537-3 was for the rats.

### 2.2. Compounds and Reagents

Ladostigil was a gift from Spero Biopharma (Jerusalem, Israel). Dulbecco’s Modified Eagle Medium/Nutrient Mixture F-12 (DMEM/F12) was used. Gentamycin sulfate and L-Glutamine were obtained from Biological Industries (Beit-Haemek, Israel) and BzATP, bovine serum albumin (BSA), and LPS were from Escherichia coli 055:B5, purified by trichloracetic acid extraction from Sigma-Aldrich Israel Ltd. (Rechovot, Israel).

### 2.3. Preparation of Microglia

Primary microglia were isolated from the brains of neonatal Balb/C mice, as previously described [30]. The brains were stripped of their meninges and enzymatically dissociated. Cells were plated on Poly-L-lysine-coated flasks for one week, re-plated for 1 to 2 h on bacteriological plates, and non-adherent cells were washed away. Microglia were propagated by incubation in 20% of the medium, and conditioned by the L-cell line that produces mouse-CSF. They were identified by morphology and positive immune reactivity to P2y12, F4/80, complement receptor-3, and Galectin-3/MAC-2 [30]. 

### 2.4. Measurement of Cytokines

Cytokines were measured as previously described [31] using ELISA Max deluxe sets (Biolegend, San Diego, CA, USA) for secreted TNFα and IL6 proteins and ELISA DuoSet (R&D Systems, Minneapolis, MN, USA) for secreted and cell-associated IL1β, according to the manufacturer’s instructions. BSA (0.1%) was used to provide the necessary protein in place of the fetal calf serum used in our previous experiments [23], which was shown to contain substances that can inhibit cytokine release [32]. We also ascertained that the concentration of LPS (0.75 µg/mL) given together with BzATP did not affect cell viability after 3 and 24 h using the MTT assay described by Denizot and Lang [33].

The effect of ladostigil on the secretion of cytokine proteins was measured at concentrations of 1 × 10^−13^ M to 1 × 10^−9^ M by adding it with BSA to microglia for 2 h before LPS (0.75 µg/mL) and BzATP (400 µM). Other microglia were treated similarly with the steroid budesonide as a reference standard at concentrations of 1 × 10^−13^–1 × 10^−11^ M. Measurements of cytokine secretion were made 8 h after the addition of LPS and BzATP. Since the levels of secreted IL1βTNF were still low, samples were concentrated 2- to 4-fold by an Amicon ultra-centrifugal filter device (Merck-Millipore, Tullagreen, Carrigtwohill, Co Cork, Ireland). ELISA was used to quantify the levels of cytokine proteins. Protein content in the microglia lysate was measured by Bradford, and levels of cytokines were calculated and presented as pg/μg of microglial protein. Each concentration of ladostigil was tested in 18–30 replicates for TNFα and IL6 and 13–18 replicates for IL1β.

### 2.5. Measurement of Ladostigil in Rat Plasma

Measurements were made in 6 male Wistar rats aged 22 months weighing 720–790 gm in which ladostigil (1 mg/kg/day) had been administered for 6 months in the drinking water, which prevented the development of learning and memory deficits [24]. The rats were lightly restrained while blood samples (0.2 μL) were taken from the tail vein between 09:00 and 12:00 into heparinized Eppendorf tubes. At the end of the experiment, blood samples were also taken by cardiac puncture from some of the rats after terminal anesthesia. They were centrifuged at 4 °C and 20,800 g for 10 min, and the plasma was stored at −80 °C until analysis by liquid chromatography–mass spectroscopy analysis. After precipitating plasma proteins with methanol, ladostigil was detected by an AB Sciex (Framingham, MA, USA) Triple Quad™ 5500 mass spectrometer in positive ion mode by electrospray ionization and a multiple reaction monitoring mode of acquisition using rivastigmine hemitartarate as an internal standard, as described in Moradov et al. [34].

### 2.6. RNA-Seq of Microglia

Ladostigil (1 × 10^−10^ M) was added to microglia for 2 h before BzATP/LPS, as described above. Cells were harvested before and 8 h after the addition of BzATP/LPS. Total RNA was extracted using the RNeasy Plus Universal Mini Kit (QIAGEN), according to the manufacturer’s protocol. Total RNA samples (1 μg RNA) were enriched for mRNAs by pull-down of poly (A). Libraries were prepared using a KAPA Stranded mRNA-Seq Kit, according to the manufacturer’s protocol, and sequenced using Illumina NextSeq 500 to generate 85 bp single-end reads (a total of 25–30 million reads per sample).

#### 2.6.1. Bioinformatic Analysis

Next-generation sequencing data underwent quality control using FastQC, version 0.11.9 (accessed on 15 March 2021). They were then preprocessed using Trimmomatic [35] and aligned to the reference genome GRCm38 with the STAR aligner [36] using default parameters. Genomic loci were annotated using GENCODE version M25 [37]. Genes with low expression were filtered out of the dataset by setting a threshold of a minimum of two counts per million in three samples.

#### 2.6.2. Gene Module Classification

Pair-wise differential analyses were performed on all three BzATP/LPS time points, and genes with an FDR < 0.01 were considered. Only the genes with an absolute log fold-change of >0.5 across two consecutive time points were labeled up- or downregulated.

### 2.7. Immunocytochemistry

For immunocytochemistry, microglia cells were plated on 12 mm round glass coverslips in 24-well sterile plates (NUNC A/S, Roskilde, Denmark) in DMEM and low glu/10% FCS-HI. Non-adherent cells were washed out after 3–4 h. Adherent microglia were incubated overnight in 0.1% BSA/DMEM/F12 and then used in experiments identical to those carried out for testing cytokine secretion. To study the expression of TNF alpha-induced protein 3 (TNFaIP3, A20) protein, ladostigil (1 × 10^−10^ M) was added to the microglia for 2 h before BzATP/LPS, and measurements were made after 8 h. TNFaIP3 was visualized by immunofluorescence confocal microscopy (Zeiss Confocal LSM 980) using an antibody against TNFaIP3 (A20; Abcam # 92324). Microglia were fixed for 15 min in 4% methanol-free formaldehyde, permeabilized for 10 min in 0.1% Triton X100, and blocked for 1 h in 10% FCS in PBS. Anti-TNFaIP3 Ab (diluted 1/200 in PBS/FCS) was added to the microglia in wet chambers overnight at 4 °C. Cy3-labeled secondary Ab goat and anti-rabbit (in PBS/FCS) were applied for 1 h followed by Alexa Fluor 488phalloidin and Dapi staining. Randomly sampled microglia were scanned by a confocal microscope at one plane that ran through the middle of their nuclei. Immunofluorescence levels in the cytoplasm of Cy3-labeled TNFaIP3 were determined by IMARIS software, Version 10.1. Optical slices of cells, 1 μm thick, were scanned sequentially and used to produce the shown maximal intensity projection images (Zeiss Zen 3.3 software). We estimated the concentration of TNFaIP3 by determining the intensity/unit area to take into account any differences among the microglia in the volume of their cytoplasm. By sampling all the cells in the same plane that runs through the center of the cell nucleus, we neutralized any preferential localization of TNFaIP3 within the cell.

To study the expression of early growth response (EGR) 1 protein, the same protocol was used as TNFAIP3 protein with some modifications. Ladostigil (1 × 10^−10^ M) was added to microglia for 2 h before BzATP/LPS, and measurements were made after 3 h. EGR1 protein (red) was visualized by immunofluorescence microscopy using a monoclonal antibody against Egr1 (cell signaling, #4153). Randomly selected low-power fields that were scanned in the same plane that runs through the center of cell nuclei were used to determine the percentage of microglia that displayed positive EGR1 protein immunoreactivity in their nuclei.

### 2.8. Statistics

The cytokine quantification data were analyzed in samples of at least 24 replicates using one-way analysis of variance (ANOVA) by IBM SPSS Statistics Version 25 followed by Duncan’s post hoc test. The assumption of the homogeneity of variances was verified using the Brown–Forsythe test for equality of group variances. Comparing two experimental groups, a two-sample *t*-test was performed. Results from experiments on cell viability, cytokine secretion from microglia, and TNFaIP3 in microglial cytoplasm are presented as mean ± SEM. Plasma levels of ladostigil are presented as mean ± SD. *p*-values of < 0.05 were considered statistically significant. Measures of TNFAIP3 in microglia were analyzed by a Kruskal–Wallis non-parametric test, and Egr1 in microglial nuclei was analyzed by Dunnett’s multiple comparison test. Principal component analysis (PCA) was performed using the R-base function “prcomp”. EdgeR was used to perform RNA read counts by the trimmed mean of the M-values normalization of RNA (TMM) and differential expression analysis [38]. Gene-set and KEGG pathway enrichment analyses were performed using the “goana” and “kegga” functions (respectively) in the “limma” R package [39]. Figures were generated using the ggplot2 R package.

## 3. Results

### 3.1. Ladostigil Concentration in the Plasma of Old Rats

The mean (±SD) plasma concentration of ladostigil in samples taken from six rats after they had been given ladostigil (1 mg/kg/day) in the drinking fluid for six months was 2.39 ± 1.08 ng/mL (8.75 ± 3.95 nM).

### 3.2. Effect of Ladostigil on Cytokine Release from Activated Microglia

The effect of BSA (0.1%) with LPS 0.75 µg/mL and BzATP (400 µM) on cell viability in arbitrary units after 3 h was 0.026 ± 0.002 and 0.029 ± 0.004 after 24 h. It did not differ from BSA, which was 0.025 ± 0.002 at both time points. The lowest concentration of ladostigil tested in microglia that significantly decreased cytokine secretion induced by BzATP/LPS was 1 × 10^−13^ M for TNFα and IL6, and 1 × 10^−12^ M for IL1β. Maximal reductions of ≈50% for IL6 and IL1β were obtained by ladostigil (1 × 10^−11^–1 × 10^−9^ M). At all concentrations of ladostigil and budesonide tested, the reductions of IL6 were greater than TNFα, (*p* < 0.001). Reductions of IL1β by ladostigil (1 × 10^−9^ M) and budesonide (1 × 10^−11^ M) were also greater than TNFα (Figure 1). The greater effect of ladostigil on the release of IL6 than TNFα, which is also seen after budesonide, may be due to the differential regulation of these cytokines by EGR1 [40].

### 3.3. Effect of Ladostigil on Genes in Microglia Assessed by RNA-seq

The reproducibility of the normalized RNA-seq read counts was assessed by performing PCA analysis for each biological sample. The biological replicates clustered tightly together, confirming the low variability within each experimental group. The sample variability (within and between groups) is illustrated by the first two principal components that comprise >83% of the variation. Only four DE genes were affected by ladostigil treatment in resting, unstimulated microglia (Figure 2A), but the expression of 25 genes was significantly altered 8 h after their activation by BzATP/LPS (Figure 2B) when ladostigil produced its inhibitory effect on cytokine secretion.

Among these were early-growth response proteins 1 and 2 (Egr1 and Egr2), matrix metalloproteinase (Mmp), Mmp 12, the tissue inhibitor of metalloprotease 1 (Timp1), and platelet-derived growth factor β (Pdgf-β) which were all downregulated by ladostigil. TNFaIP3 was upregulated (Figure 3A,B).

The connected genes with a STRING score of >0.6 are shown in Figure 4. The network connectivity is highly significant (*p*-value: 2.2 × 10^−4^).

### 3.4. Ladostigil Treatment Increases TNFaIP3 Protein in BzATP/LPS-Activated Microglia

The addition of BzATP/LPS to microglia caused a small but significant increase in TNFaIP3 protein in the cytoplasm (*p* < 0.05). This was increased further (*p* < 0.001) by the addition of ladostigil (1 × 10^−10^ M) (Figure 5 and Table 1).

### 3.5. Ladostigil Treatment Decreases EGR1 Protein in the Nucleus of BzATP/LPS-Activated Microglia

The addition of BzATP/LPS to microglia significantly increased the number of microglia containing EGR1 protein in their nuclei (*p* < 0.001). This was decreased significantly (*p* < 0.001) by ladostigil (Figure 6 and Table 2).

## 4. Discussion

The dysfunction of mitochondria and the generation of reactive oxygen species (ROS) occur in the aging brain and are early contributory events to neurodegeneration and Alzheimer’s disease (AD) [6,7]. ROS releases ATP [41], which activates purinergic A_2A_ receptors (A_2A_R) on the microglia membrane, causing them to retract their processes [42]. A_2A_R and P2x7 receptors (P2x7R) are upregulated in the brains of patients with AD [43] and in the hippocampus of aging rats with memory impairment. The stimulation of A_2A_R in the brain by ATP further increases neurodegeneration [44], while the activation of P2x7R on microglia releases several pro-inflammatory cytokines [28]. In the current study, the addition of BzATP to activate P2x7R in microglial cultures in addition to LPS enabled the measurement of IL1β protein secretion. Both IL1β and IL6 decreased by ≈ 50% at concentrations of ladostigil of 0.01–1 nM.

Chronic treatment with ladostigil (1 mg/kg/day) in aging rats prevented the upregulation of A_2A_R and memory decline [25]. The mean plasma concentration of ladostigil in these rats was found to be 2.39 ± 1.08 ng/mL or 8.75 ± 3.95 nM. However, ladostigil was not measured in the brain of the aging rats, and peak concentrations in the cerebral cortex in young adult rats after acute oral administration of 5 mg/kg were ≈25% of those in plasma (unpublished observations). Assuming a similar plasma-cortical ratio after the chronic administration of 1 mg/kg/day in aged rats, a concentration of ≈2.2 nM is obtained. After s.c. injection of 5 mg/kg in mice, the concentration of ladostigil in the brain was also 25–50% of that in plasma [34]. Since a dose of 1 mg/kg/day also decreased the gene expression of TNFα, IL6, and IL1β in the brain [23], it is reasonable to assume that there could have been enough ladostigil in the brain of the aging rats to have reduced cytokine release.

Cytokine secretion from microglia activated by LPS and BzATP was accompanied by a significant upregulation of transcription factors, Egr1, Egr2, and PDGF-β. Egr1 and Egr2 are downstream signaling targets of P2x7R [40] that increase in LPS-activated mixed astrocyte–microglial cultures [45]. Egr1 is rapidly and transiently induced in different cell types in response to a variety of stimuli, including oxidative stress, radiation injury, electrical stimulation, and neurotransmitter activity. It is activated by intracellular pathways, including MAPKs, ERK, and p38. Egr1/Krox24 gene expression in the hippocampus was shown to be related to the severity of AD in human subjects [46]. Egr1 also accelerates tau phosphorylation and the processing of amyloid precursor protein to β-amyloid in a mouse model of AD [47]. Much less is known about the role of Egr2 as a mediator of inflammation.

PDGF-β is released in the cerebral spinal fluid (CSF) from pericytes and is a specific marker for pericyte injury associated with a loss of integrity in the blood–brain barrier [48], which declines in normal aging and more rapidly in AD. A correlation was found between age and PDGF-β in CSF, with the highest levels found in subjects with mild cognitive impairment (MCI) and AD [49].

Ladostigil decreased the expression of Egr1 and Egr2 transcripts in BzATP/LPS-activated microglia and Mmps 12, all of which regulate cytokine release [50], and it also reduced the expression of PDGF-β. It significantly reduced the amount of EGR1 protein in the nucleus three h after it had been elevated by BzATP/LPS. On the other hand, ladostigil upregulated the gene expression of the ubiquitin-modifying enzyme, TNFaIP3, and increased the levels of this protein in the microglial cytoplasm. TNFaIP3 terminates the activation of NFκB in response to stimulation by LPS, IL1β, TNFα, IL6, or CD40 [51]. TNFaIP3 prevents the NFκB-dependent upregulation of NLRP3 and conversion of pro-IL1β to mature IL1β through the binding of its A20-like zinc finger domain to ubiquitin chains [52]. It also blocks IKKα/β activation by the upstream kinase, Tak1 [53]. Moreover, the brains of mice lacking TNFaIP3 have a larger number of microglia with shorter and fewer processes, resembling those after chronic infection or aging [54]. Together, these observations suggest that the elevation of TNFaIP3 could protect the organism against inflammatory conditions occurring in the aged brain [19].

An increase in cytosolic TNFaIP3 by ladostigil via the alteration of various feedback-controlling mechanisms [55] could be responsible for the reduction in the phosphorylation of ERK and/or p38 and the decrease in nuclear EGR1. This, in turn, explains how ladostigil reduced the formation and secretion of cytokines in BzATP/LPS-activated microglia in the current study. Restoring the aberrant signaling of these genes and their proteins by ladostigil to normal enables us to explain how they prevented the morphological and inflammatory changes in the brain regions of aging rats [25] and the attenuation of the decline in memory in the whole brain and hippocampal volumes in elderly subjects with MCI [56].

## Figures and Tables

**Figure 1 biomolecules-14-00112-f001:**
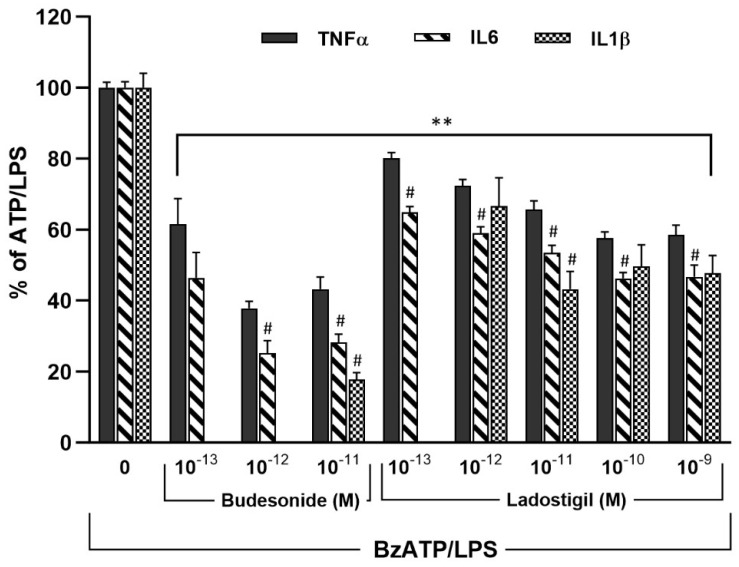
A dose–related reduction in the release of TNFα, IL6, and IL1β from microglia activated by BzATP/LPS. Ladostigil or budesonide was added 2 h before BzATP and LPS in the presence of 0.1% BSA. ANOVA for TNFα; F_5,275_ = 85.4, *p* < 0.0001; IL6; F_5,282_ = 141.7, *p* < 0.0001; IL1β; F_4,73_ = 18.14, *p* < 0.0001. All concentrations of ladostigil and budesonide tested reduced the three cytokines; ** *p* < 0.01. This was significantly different from the value for TNFα; # *p* < 0.05.

**Figure 2 biomolecules-14-00112-f002:**
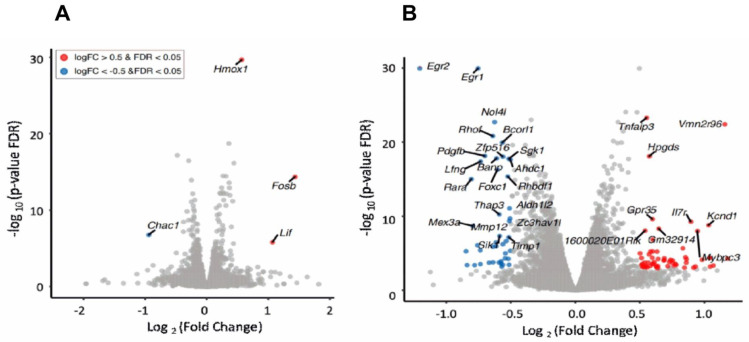
Ladostigil treatment alters gene expression in BzATP/LPS-activated microglia. (**A**) Volcano plots showing the log-fold change vs. −log10 (*p*-value), as calculated by edgeR, and differential expression analysis in ladostigil-treated microglia compared to untreated cells. (**B**) Volcano plots showing the log-fold change vs. −log10 (*p*-value) of ladostigil-treated microglia 8 h after the addition of BzATP/LPS. Blue dots indicate genes that are downregulated and red dots indicate genes that are upregulated by ladostigil. Cut-off value, ±0.5.

**Figure 3 biomolecules-14-00112-f003:**
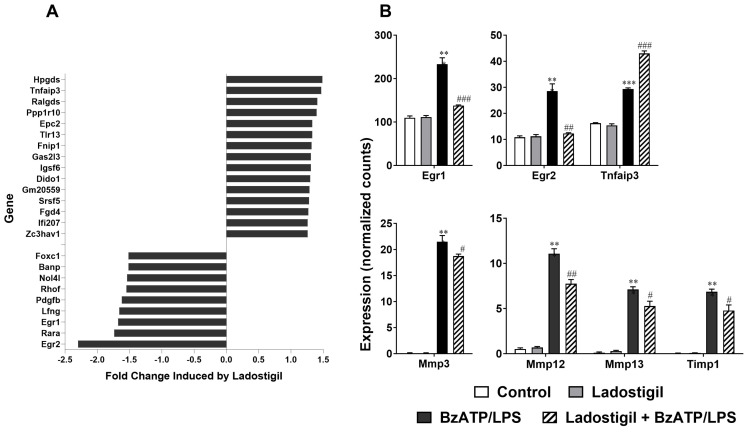
Genes altered by ladostigil treatment. (**A**) Bars show the change caused by ladostigil in gene expression in microglia after BzATP/LPS-induced activation. Bars on the right show genes that were upregulated and bars on the left show genes that were downregulated. Cut-off value, ±0.5. (**B**) Selected, differentially expressed genes after BzATP/LPS treatment, with or without ladostigil. Significantly different from the unstimulated control, ** *p* < 0.01, *** *p* < 0.001; significant effect of ladostigil, # *p* < 0.05; ## *p* < 0.01 ### *p* < 0.001.

**Figure 4 biomolecules-14-00112-f004:**
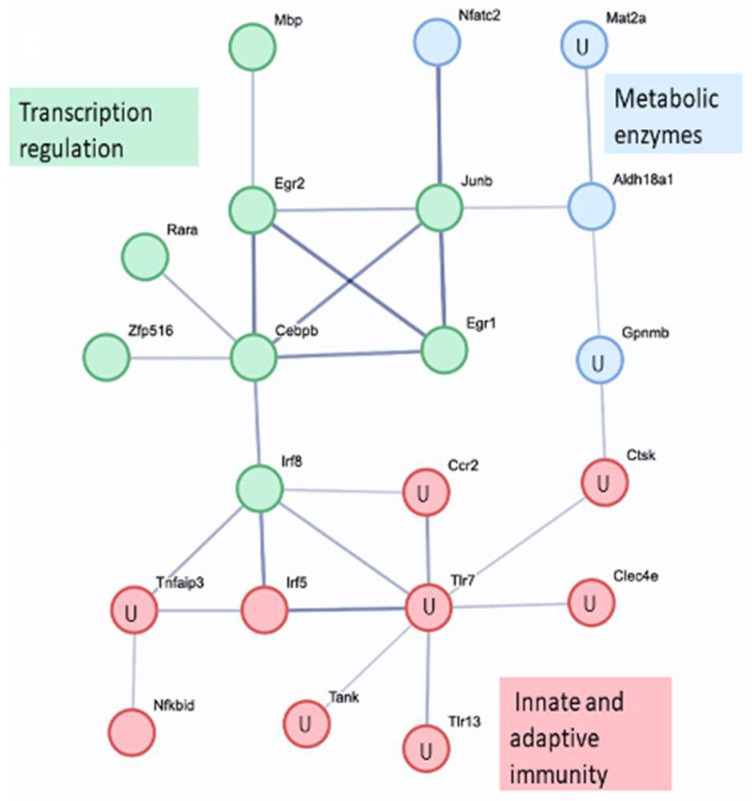
STRING protein–protein interaction network. Differentially expressed genes 8 h after BzATP/LPS activation with and without ladostigil. All clusters are labeled by their main cellular functions, and the larger connected network (red color) is partitioned into sub-clusters for functional annotation. The letter U in the node indicates upregulated genes. The other nodes are downregulated genes. STRING protein–protein interaction enrichment, *p* = 2.2 × 10^−4^.

**Figure 5 biomolecules-14-00112-f005:**
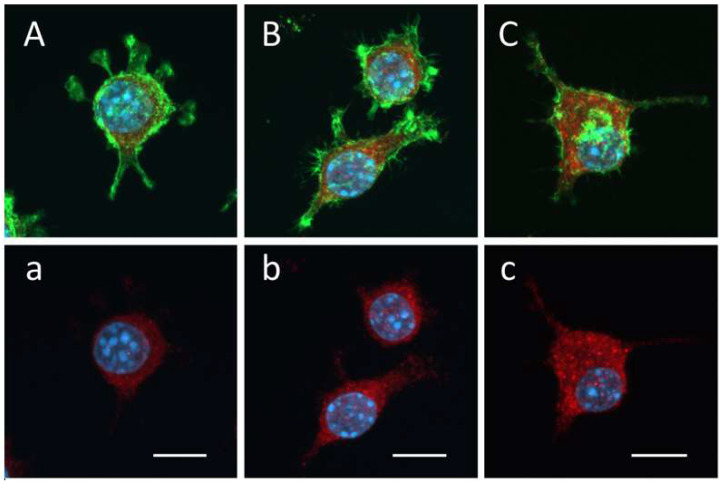
Ladostigil increases levels of TNFAIP3 protein immunoreactivity in microglial cytoplasm in the presence of BzATP/LPS. Three representative high-power immunofluorescence confocal microscopy images of single microglia are displayed: (**A**,**a**), medium with 0.1% BSA; (**B**,**b**) medium plus BzATP/LPS; and (**C**,**c**) medium plus ladostigil + BzATP/LPS. TNFAIP3 protein was visualized (red) by immunocytochemistry using an antibody against TNFAIP3 protein, F-actin (green) by Alexa 488-labeled phalloidin, and nuclei (blue) by DAPI staining. Optical slices of phagocytes, 1 μm thick, were scanned sequentially and used to produce the shown maximal intensity projection images (Zeiss Zen 3.3 software). TNFAIP3 protein immunoreactivity is detected in the cytoplasm but not the nuclei (**a**–**c**). In fields (**A**–**C**), TNFAIP3 protein-positive immunoreactivity, F-actin and nuclei are displayed. In fields (**a**–**c**), only TNFAIP3 protein-positive immunoreactivity and nuclei are displayed. Calibration bars: 10 µm.

**Figure 6 biomolecules-14-00112-f006:**
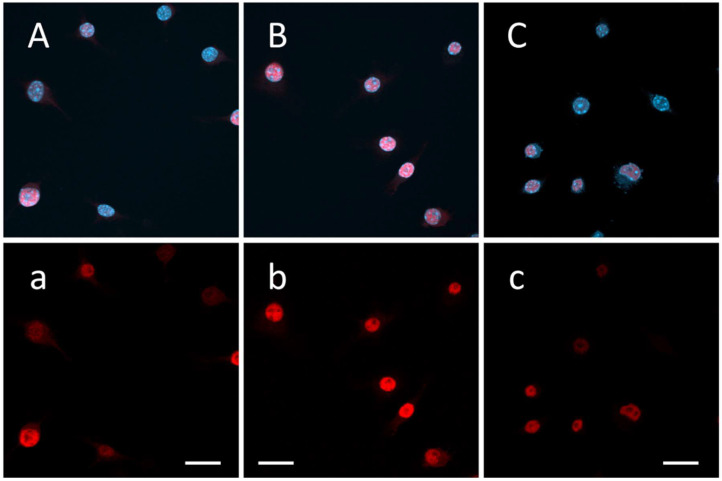
Ladostigil decreases the number of microglia displaying positive EGR1 protein immunoreactivity in their nuclei induced by BzATP/LPS treatment. Three representative low-power immunofluorescence confocal microscopy fields are displayed: (**A**,**a**) medium with 0.1% BSA; (**B**,**b**) medium plus BzATP/LPS; (**C**,**c**) ladostigil + BzATP/LPS. Egr1 was visualized using immunocytochemistry and a monoclonal antibody against EGR1; nuclei (blue) were visualized using DAPI staining, and EGR1 (red), overlaying nuclei appear pink. In fields (**A**–**C**), both EGR1 protein-positive immunoreactivity and nuclei are displayed. In fields (**a**–**c**), only EGR1 protein-positive immunoreactivity is displayed. Calibration bars: 20 µm.

**Table 1 biomolecules-14-00112-t001:** Quantification of TNFAIP3 protein in the microglial cytoplasm.

Treatment	N	Mean Fluorescence Intensity ± SEM (AU/µm^2^)
Medium	32	19.8 ± 0.6
BzATP/LPS	53	23.3 ± 1.0 *
Ladostigil + BzATP/LPS	58	28.8 ± 0.9 ***^###^

Randomly selected fields (as those shown in Figure 5) were used. N = number of fields sampled. Significance of difference by ANOVA, *p* < 0.0001, and Bonferroni’s multiple comparison test; medium vs. BzATP/LPS * *p* < 0.05 and medium vs. ladostigil + BzATP/LPS, *** *p* < 0.001; BzATP/LPS vs. ladostigil + BzATP/LPS, ^###^ *p* < 0.001.

**Table 2 biomolecules-14-00112-t002:** Quantification of EGR1 protein immunoreactivity in microglial nuclei.

Treatment	N	Mean Percent of Nuclei with Positive EGR1 Protein Immunoreactivity ± SEM
Medium	32	19.0 ± 4.2
BzATP/LPS	53	82.3 ± 3.7 ***
Ladostigil + BzATP/LPS	58	56.0 ± 3.9 ***^###^

Randomly selected fields (as those shown in Figure 5) were used. N = number of fields sampled. Significance of difference by ANOVA, *p* < 0.0001, and Bonferroni’s multiple comparison test; medium vs. BzATP/LPS and medium vs. ladostigil + BzATP/LPS, *** *p* < 0.001; BzATP/LPS vs. ladostigil + BzATP/LPS, ^###^ *p* < 0.001.

## Data Availability

RNA-seq data files were deposited in ArrayExpress under accession E-MTAB-10450.

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
