# Peer review of "Ladostigil Reduces the Adenoside Triphosphate/Lipopolysaccharide-Induced Secretion of Pro-Inflammatory Cytokines from Microglia and Modulate-Immune Regulators, TNFAIP3, and EGR1"

_biomolecules, 2024, doi:10.3390/biom14010112_

Round 1

Reviewer 1 Report

Comments and Suggestions for Authors

In this manuscript, the authors explore the mechanism by which ladostigil reduces cytokine release from microglia. Through RNA sequencing and immunocytochemistry, they discovered that ladostigil significantly diminishes Egr1 gene expression and lowers its protein levels in the nucleus. Furthermore, ladostigil elevates the levels of TNF alpha-induced protein 3 (TNFaIP3), which inhibits cytokine release in microglia. In addition, the plasma concentration of ladostigil in rats treated with the drug was determined. I have some comments as follows:

1. Given that the authors have rats treated with ladostigil, it would be beneficial to measure the levels of Egr1 and TNFaIP3, and compare if they are consistently changed in relation to their in vitro data.

2. It’s important to note that in vitro microglia differ from in vivo microglia. The isolation process often modifies the homeostatic gene expression of microglia. The immune activation of in vitro microglia changes post-isolation, and their response can differ based on the isolation methods used. Therefore, it’s inappropriate to conclude whether there was sufficient ladostigil in the brains of the aging rats to reduce cytokine release based solely on these in vitro data (lines 342-344). The authors should discuss this more cautiously and acknowledge the limitations of their in vitro studies.

Comments on the Quality of English Language

Authors shall mind their English writing as there are several mistakes. 

Author Response

Given that the authors have rats treated with ladostigil, it would be beneficial to measure the levels of Egr1 and TNFaIP3, and compare if they are consistently changed in relation to their in vitro data.

In our study (Lineal et al, 2020) we found marked differences in the genes in four different brain areas associated with memory and in their response to aging and ladostigil. In the perirhinal and frontal cortices, aging decreased the expression of Egr1 and Egr2 compared to that adult rats and this was further decreased by ladostigil (see Supplements 2 and 3).

  1. It’s important to note that in vitro microglia differ from in vivo microglia. The isolation process often modifies the homeostatic gene expression of microglia. The immune activation of in vitro microglia changes post-isolation, and their response can differ based on the isolation methods used. Therefore, it’s inappropriate to conclude whether there was sufficient ladostigil in the brains of the aging rats to reduce cytokine release based solely on these in vitro data (lines 342-344). The authors should discuss this more cautiously and acknowledge the limitations of their in vitro studies.

The reviewer is correct. However, we showed that this dose and treatment regimen of ladostigil, also significantly decreased the mRNA of the same pro-inflammatory cytokines in the brain of aging rats (Panarsky et al., 2012). This supports the suggestion that there could have been enough drug in the brain to have effected cytokine release. We have added this information and stated our findings more cautiously as suggested.

Reviewer 2 Report

Comments and Suggestions for Authors

The manuscript sent by Reichert et al. is novel and interesting. The introduction provides sufficient and appropriate information on the topic. The methods are relevant and adequately described. The results support the conclusions. Its publication is adequate, with some minor changes.

My observations are:

Results. Fig 1. I don't understand the point of comparing the levels of IL-6 and IL1β with those of TNFα

Discussion. I think it would be worth delving deeper into the effect of Ladostigil on gene expression.

Perhaps the following bibliography could enrich the discussion:

Yogev-Falach et al., A multifunctional, neuroprotective drug, ladostigil (TV3326), regulates holo-APP translation and processing. FASEB J. 2006 Oct;20(12):2177-9. doi: 10.1096/fj.05-4910fje. 

Bar-Am et al., The novel cholinesterase-monoamine oxidase inhibitor and antioxidant, ladostigil, confers neuroprotection in neuroblastoma cells and aged rats. J Mol Neurosci. 2009 Feb;37(2):135-45. doi: 10.1007/s12031-008-9139-6. 

Weinreb et al., The application of proteomics and genomics to the study of age-related neurodegeneration and neuroprotection. Antioxid Redox Signal. 2007 Feb;9(2):169-79. doi:10.1089/ars.2007.9.169. 

Author Response

The manuscript sent by Reichert et al. is novel and interesting. The introduction provides sufficient and appropriate information on the topic. The methods are relevant and adequately described. The results support the conclusions. Its publication is adequate, with some minor changes.

My observations are:

Results. Fig 1. I don't understand the point of comparing the levels of IL-6 and IL1β with those of TNFα

Discussion. I think it would be worth delving deeper into the effect of Ladostigil on gene expression.

Perhaps the following bibliography could enrich the discussion:

Yogev-Falach et al., A multifunctional, neuroprotective drug, ladostigil (TV3326), regulates holo-APP translation and processing. FASEB J. 2006 Oct;20(12):2177-9. doi: 10.1096/fj.05-4910fje. 

Bar-Am et al., The novel cholinesterase-monoamine oxidase inhibitor and antioxidant, ladostigil, confers neuroprotection in neuroblastoma cells and aged rats. J Mol Neurosci. 2009 Feb;37(2):135-45. doi: 10.1007/s12031-008-9139-6. 

Weinreb et al., The application of proteomics and genomics to the study of age-related neurodegeneration and neuroprotection. Antioxid Redox Signal. 2007 Feb;9(2):169-79. doi:10.1089/ars.2007.9.169. 

Fig 1 shows that both budesonide and ladostigil cause a greater reduction in IL-6 and IL-1β than TNFα, probably because IL-6 and TNFα are regulated differently by Egr1 in microglia (Friedel et al 2011). This statement has been added in the results section close to Fig. 1.

We are well acquainted with the findings in the papers cited above. In the study of Weinreb et al. ladostigil was given by gavage, once daily for 30 days, at a dose of 1 mg/kg to aged female rats and affected some proteins associated with oxidative stress. This is an interesting finding but not really comparable to the regimen of 6 months’ administration of 1 mg/kg/day to male rats given in the drinking fluid in our studies. Moreover, the concentration of ladostigil used in the in vitro studies of Yogev-Falach et al. and BarAm et al, are at least 1,000-fold high than those we used in our microglial cultures.

Round 2

Reviewer 1 Report

Comments and Suggestions for Authors

The authors have addressed my comments. I don't have further concerns.